# Predicting Emotion with Biosignals: A Comparison of Classification and Regression Models for Estimating Valence and Arousal Level Using Wearable Sensors

**DOI:** 10.3390/s23031598

**Published:** 2023-02-01

**Authors:** Pekka Siirtola, Satu Tamminen, Gunjan Chandra, Anusha Ihalapathirana, Juha Röning

**Affiliations:** Biomimetics and Intelligent Systems Group, University of Oulu, P.O. Box 4500, FI-90014 Oulu, Finland

**Keywords:** emotion detection, valence, arousal, wearable sensors, regression, classification, machine learning

## Abstract

This study aims to predict emotions using biosignals collected via wrist-worn sensor and evaluate the performance of different prediction models. Two dimensions of emotions were considered: valence and arousal. The data collected by the sensor were used in conjunction with target values obtained from questionnaires. A variety of classification and regression models were compared, including Long Short-Term Memory (LSTM) models. Additionally, the effects of different normalization methods and the impact of using different sensors were studied, and the way in which the results differed between the study subjects was analyzed. The results revealed that regression models generally performed better than classification models, with LSTM regression models achieving the best results. The normalization method called baseline reduction was found to be the most effective, and when used with an LSTM-based regression model it achieved high accuracy in detecting valence (mean square error = 0.43 and $R^{2}$-score = 0.71) and arousal (mean square error = 0.59 and $R^{2}$-score = 0.81). Moreover, it was found that even if all biosignals were not used in the training phase, reliable models could be obtained; in fact, for certain study subjects the best results were obtained using only a few of the sensors.

## 1. Introduction

Wearable wrist-worn sensors are commonly used to monitor human motion based on inertial sensors such as accelerometers, gyroscopes, and and magnetometers. In addition, wearables can include sensors for measuring biosignals. Nowadays, wrist-worn wearable devices can house a wide range of biosensors, including photoplethysmography (BVP) to measure the blood volume pulse, heart rate (HR), and heart rate variability (HRV), thermometers (ST) to measure skin temperature, and electrodermal activity (EDA) sensors to measure galvanic skin responses. Based on these, it is possible to monitor human motion along with monitor other aspects of human behavior and events occurring inside the human body.

Articles have shown good results in detecting stress and affect states based on the data provided by wearable sensors. For instance, in [1], eight affect states (excited, happy, calm, tired, bored, sad, stressed, and angry) were detected based on acceleration, electrocardiogram, blood volume pulse, and body temperature signals. The results were promising, especially when personal models were used in the recognition process. Similarly, in [2], heart rate, blood volume pulse, and skin conductance were used to detect seven affective states (fun, challenge, boredom, frustration, excitement, anxiety, and relaxation); using artificial neural networks, most of these could be detected with accuracy over 80%. In [3], a classifier able to detect high and low stress as well as non-stressful situations in laboratory conditions was developed based on wearable wrist-work sensors. The results showed that the two stress classes could be detected with an accuracy of 83%. In [4], a binary classifier was trained to detect stress and non-stressed state, and it was noted that stress could be detected using the sensors in commercial smartwatches. There have been several other studies showing that stress detection based on classification models can be performed with high accuracy using user-independent models; see for instance [5,6].

In the past, automatic emotion recognition from biosensor data has focused on detecting discrete classes of emotion. However, humans have hundreds of emotions, and id discrete classes can be recognized it is possible to recognize a limited number of them. In addition, it is important both to recognize the affected state and to recognize the level of the affected state, such as a person being slightly happy, extremely happy, or anything between these two. In fact, psychological studies have suggested that the full spectrum of human emotion can be characterized by just a few dimensions. One common strategy to express human emotion in discrete classes is to divide emotions into valence and arousal ([7,8]). Valence is the horizontal extent, ranging from displeasure to pleasure, and arousal is the vertical extent, ranging from deactivation to activation. By combining valence and arousal, every human emotion can be expressed; often, these are visualized using Russell’s circumplex model of emotions (see Figure 1).

In this article, valence and arousal levels are predicted based on biosignal data collected using wearable wrist-worn sensors. The novel contributions of our paper are as follows:It is shown that fine-grained levels of valence and arousal can be detected based on the data of wrist-worn sensors with high reliability when the data and the target values provided by the study subjects are normalized in the right way.We compare the suitability of different classification and regression models for detecting the levels of valence and arousal.We explore the finding that the same features and sensors may not be optimal for every individual.

The rest of this article is organized as follows. Related works are introduced in Section 2, and the data used in the experiments are explained in Section 3. Section 4 introduces the methods used in the study, and  Section 5 explains the experimental setup, the applied methods, and the obtained results. Finally, a discussion is presented in Section 6 followed by our conclusions and prospects for future work in Section 7.

## 2. Related Work

In [9], valence and arousal detection using regression models were studied based on audio data. There have been several audio- and video data-based studies, for instance, [9,10], on detecting valance and arousal based on continuous response values and regression models. However, when it comes to wearable sensors, emotion recognition has in the past focused on identifying discrete classes of emotion. There have been a number of studies in which valence and arousal levels were detected; however, in these, the detection was based on dividing valance and arousal values into discrete classes, meaning that the prediction models were based on classification methods. Often, valance and arousal values are only divided into two discrete classes, high or low arousal, for instance in [11,12,13,14]; however, there have been studies in which more fine-grained classes were studied as well.

In [15], valence and arousal were divided into three classes: low, neutral, and high arousal/valance. However, in the final classification only the low and high arousal/valence observations were used. A dataset was collected from 21 study subjects playing games with increasing difficulty and self-reporting their valence and arousal levels while playing. The participants were wearing OpenBCI headsets and JINS MEME eyewear, and these were used to collect electroencephalography (EEG), electrooculography (EOG), and kinematic motion data (acceleration from the head and glasses, and gyroscope data from the glasses). Classification methods such as ensemble learning and random forest were used as classifiers. In the study, ten-fold cross-validation was used instead of leave-one-subject-out cross-validation. The best accuracies were obtained using an ensemble learner, and in the binary case, these were 73% for arousal and 80% for valence. In [16], two datasets were studied: the publicly open CASE dataset, containing electrocardiogram (ECG), BVP, EDA, respiratory rate, ST, and EMG (electromyography) signals, and another dataset called MERCA containing data from the autonomic nervous system (HR, HRV, ST, and EDA) and oculomotor nerve system (pupil dilation, saccadic amplitude, and saccadic velocity) collected using an Empatica E4 wrist-worn sensor and wearable eye tracker. The aim of the study was to detect the levels of valence and arousal. For this purpose, three scenarios were studied: a binary case in which valence and arousal were divided into two classes (high and low), a three-class case (low, neutral, and high arousal/valance), and a four-class case (high valence + high arousal, high valence + low arousal, low valence + high arousal, and low valence + low arousal). Different machine learning and deep learning models were used in the experiments; when leave-one-subject-out cross-validation was used, the recognition rate for the binary case was around 70%, while it was lower for the cases with three and four classes.

However, as the level of valence and arousal can be high, low, or anything between these, dividing the level of valence and arousal into discrete classes is not the best option, and there is evidence that wearable sensors can be used to predict continuous affect state values as well. In [17], a significant correlation was found between valence levels and cortisol levels, which have been accepted as a reliable physical measure of emotions. In addition, it was found that EEG (electroencephalography) signals collected using a wearable device correlated with valence levels, showing that the signals of wearable devices correlates with cortisol levels. The study did not rely on machine learning methods; however, what was noticeable was that valence levels were not divided into discrete classes, meaning that the study shows that wearables, in this case EEG sensors, can be used to detect continuous valence level values.

When it comes to detecting continuous target values such as the level of valence and arousal using machine learning and artificial intelligence methods, prediction needs to be built by relying on regression models instead of classification models. However, it seems that there are not many studies where valance and arousal levels are estimated using regression models. In [18], arousal level was estimated based on ECG, respiratory rate, EDA, and ST signals using only simple a linear regression model, which is not an up-to-date regression model. Moreover, the authors did not study valence at all. On the other hand, regression has been applied to other related problems. For instance, in [19] the authors used statistical methods such as regression to predict anxiety based on wearable data (in this case BVP, ST, EDA, and microphone data), and in other studies it has been shown that continuous stress levels can be estimated based on regression and statistical methods (for instance [20,21,22,23]).

These related studies are presented in Table 1. What can be noted from the table is what is not studied, how well the data of wrist-worn wearable sensors can be used to estimate the level of both valence and arousal, and how well modern regression models can predict valence and arousal levels compared to classification models when the valence and arousal levels are divided into fine-grained discrete classes.

## 3. Experimental Dataset

The experiments in this study were carried out based on the open WESAD dataset [24], which was gathered using an Empatica E4 [25] wrist device and chest-worn RespiBAN device. In this study, only the Empatica E4 data was used. This device includes sensors to measure acceleration (ACC), skin temperature (ST), electrodermal activity (EDA), blood volume pulse (BVP), heart rate (HR), and heart rate variability (HRV).

WESAD contains data from 15 participants; from each study subject, it contains baseline data, data from a stressful situation, data from a state of amusement, and data from two meditation states. The purpose of the meditation sessions was to relax the study subject after each task, and the meditation data were not used in this study. The baselines were collected at the beginning of the data-gathering session while the participants were sitting/standing at a table and reading neutral magazines. During the gathering of stress data, participants had two tasks: (1) they had to provide a public presentation, and (2) they had to solve arithmetic tasks. Data from the state of amusement was collected while the participants were watching funny videos. The length of the stressful situation was approximately 10 min, the amused situation was 6.5 min, and the relaxed situation (baseline) was 20 min. After each task, the subjects were asked to fill in a self-report consisting of three types of questionnaires: PANAS [26], shortened STAI [27], and SAM [28]. PANAS asks whether a person had certain positive and negative moods during tasks, STAI concentrates on questioning how strong a person’s feelings of anxiety are, and SAM is used to ask about a person’s level of valance and arousal on a scale of 1–9. Therefore, when the models to predict valence and arousal levels were trained, the labels used in the training process were based on these subjective estimations. In this study, answers to the SAM questionnaire were scaled to [−4, 4] and used to define the correct target variables. This means that when classification methods are used, instances are classified into nine classes.

In the pre-processing stage, as suggested in [29], not all of the baseline data were used in the experiments. In fact, the first half of the baseline data were removed from the dataset, as it is possible that immediately after starting to gather data for the baseline the study subject’s body may not be in a relaxed state. Moreover, pre-processing of the EDA signal was carried out following the guidelines in [24]. A 5 Hz low-pass filter was applied to the raw EDA signal, then it was divided into phasic and tonic parts [30] using cvxEDA (https://github.com/lciti/cvxEDA). BVP and ST signals were used as they were.

For the model training, signals were divided into windows and features were extracted from these. A window size of 60 s was used in the experiment, which is the same as used in [4,24] for stress detection. The slide between two adjacent windows was 30 s. Different statistical features (min, max, mean, std, percentiles) per signal (BVP, ST, EDA, phasic and tonic parts of EDA), and physiological features were extracted from the HRV and PPG signal using HeartPy (https://python-heart-rate-analysis-toolkit.readthedocs.io/en/latest/). In addition, the slope was calculated from the ST. Features were not extracted from the accelerometer signal, as the accelerometer measures movement. Due to this, different activities performed during different tasks may be visible in acceleration data, leading to situations where accidental activities are detected instead of emotions. Therefore, only features extracted from biosignals were used to train the models. The full list of extracted features is shown in Table 2. These features and libraries are commonly used and recommended for extracting features and pre-processing biosignal data [31]. After the features were extracted, rows with NaN- and Inf-values were removed from the dataset.

The feature matrix was scaled to level 0–1, and is visualized in Figure 2 using t-SNE, a dimensionality reduction technique that can be used to project high-dimensional data into a low-dimensional space [32]. The figure shows that t-SNE effectively clusters data from different valance and arousal estimates reported by study subjects into own clusters, as shown in Figure 2a,b. However, the high number of clusters and wide spread of samples within the same class but different clusters makes analyzing this dataset challenging. This indicates a high level of variation within classes, and could be due to differences among study subjects. In addition, the high number of clusters indicates that the dataset has a complex underlying structure. Therefore, it may require advanced techniques in order to achieve good results. Moreover, the figures show that the dataset is highly imbalanced. Most of the samples are quite neutral, as their label is close to zero, and the dataset does not contain many extreme values. This makes the data analysis process even more challenging.

## 4. Methods

This section introduces the normalization methods compared in this study as well as the classification and regression models and the performance metrics used to compare them.

### 4.1. Normalization

People are different, and due to this, biosignals collected from the study subjects differ from individual to individual. In addition, biosignals are affected by daily changes, for instance those caused by sleep quality and chronic stress. Therefore, the difference in biosignals between the subjects can be considerable, and this may pose challenges for prediction models. Due to this, the prediction power of recognition models trained using raw data can vary a great deal between individuals.

Data normalization is often found to be an effective method to remove participant-specific effects on the data, such as daily changes and different natural ranges, and is a good way to make trained models more generalized to any study subject [33]. Due to this, normalization is a powerful method to adapt models to the current status of the study subject’s body as well as to the calibration status of the sensor itself. Moreover, by regularly calculating the required parameters for normalization, for instance, every morning, normalization can be used as a tool to adapt models to the changes happening inside the human body, which affects the sensor readings as well. In this study, four datasets were created in order to experiment with the effects of different normalization methods. Figure 3 explains how these were created.

Person-specific *z*-score normalization has been found to be the most effective way to normalize biosignals [33,34] when classifying affect and stress stages. Due to this, it was used in this study. The *z*-score normalized value $z_{i}$ for observation $x_{i}$ can be calculated using the equation
(1)$$z_{i}=\frac{x_{i}-\mu }{\sigma },$$
where the μ is mean and the σ is standard deviation calculated from the whole signal $X,x_{i}\in X$ collected from an individual. This normalization is performed separately for each collected biosignal (EDA, BVP, HRV, and ST).

Baseline reduction is commonly used as a normalization method when dealing with data on emotions. Individual valence and arousal value estimations for the baseline data were reduced from all the target values for valence and arousal; thus, the target value for each individual for valence and arousal at the baseline was zero, as the baseline is considered a neutral stage. Because of the normalization, subject-wise differences from the target values could be removed.

The third normalization method tested in this study was a combination of the first two approaches; *z*-score normalization was used to remove individual differences from the signals, then baseline reduction was used to remove individual differences from the labels.

To determine the benefits of normalization, these three normalization methods were compared to situations where signals and labels were not normalized and the models were instead trained based on raw data which was not normalized at all.

It should be noted that baseline reduction has an effect on the number of target variables. When raw questionnaire data were used in the modeling process, the number of classes for valence level prediction was seven (none of the study subjects reported valence levels −3 and 4), and for arousal level prediction it was nine. However, after baseline reduction the number of classes for valence level prediction was eight and for arousal level prediction it was six.

### 4.2. Prediction Methods

As one of the purposes of our study was to compare classification and regression methods, several different classification and regression methods were assessed. Long Short-Term Memory (LSTM) is a variant of recurrent neural networks; it is highly suitable for time-series prediction, as it is capable of learning long-term dependencies from the data [35]. In this study, one hidden layer was used, as it has been shown in [36] that an LSTM with one layer provides better results than one with two layers when studying wearable sensor data. The LSTM layer of the model used in this study had 64 units, and the model had around 17,000 parameters to train. AdaBoost [37], Random Forest [38], XGBoost (eXtreme Gradient Boosting) [39], and Histogram-GBM (inspired by LightGBM [40]) are ensemble methods that train a group of weak learners, usually decision trees, and make a final prediction that is a combination of these. In comparison to decision trees, linear regression, and LDA (linear discriminant analysis), the latter are simpler methods. In this article, LSTM, AdaBoost, Random Forest, and XGBoost were used for both classification and regression, LDA and decision tree were used for classification only, and linear regression and Histogram-GBM were used for regression only.

### 4.3. Performance Metrics

All regression models were evaluated using two different types of evaluation parameters commonly used with regression models, namely, $R^{2}$ and the mean squared error (MSE) [41]. In addition, certain models were evaluated in greater detail using the root mean squared error (RMSE) and mean average error (MAE) [41]. Normally, the results of classification models are evaluated using a confusion matrix and performance metrics calculated from it; because in this case the classification classes were fine-grained and ordinal, the performance of the classification methods was evaluated using $R^{2}$ and MSE as well. When analyzing model performance using these metrics, it is important to note that a value of zero is optimal for MSE, RMSE, and MAE, while a value of one is optimal for $R^{2}$.

Traditionally, the performance of classification methods is analyzed using performance metrics such as accuracy, sensitivity, specificity, etc. However, as in this article the idea is to compare classification and regression methods and regression models cannot be analyzed using these metrics, both classification and regression models were analyzed using MSE, RMSE, $R^{2}$, and MAE. In fact, as valence and arousal are continuous phenomena and the targets for them are ordinal, it is natural that all the models be analyzed using these metrics. Moreover, evaluating both classification and regression models using the same performance metrics makes their comparison easier.

## 5. Experimental Setup and Results

The results of the experiments are presented in this section. All the results were calculated using the leave-one-subject-out method, meaning that one study subject’s data is used for testing and all the other data is used for training, with the process then repeated in turn (Figure 4). Due to this, the trained models are user-independent. When the results for all the study subjects were obtained, they were combined as one sequence and MSE and $R^{2}$ values were calculated from these combined sequences. As most of the models used in this article contain random elements, the models were trained five times. All of the results presented in this section are averages from these runs, with the standard deviation between the runs shown in parenthesis. The scale of the target variables was [−4, 4]; if the estimated value was outside of this scale, it was replaced with −4 or 4.

### 5.1. Comparison of Prediction Models and Normalization Methods

The results presented in Table 3 show how well different classification and regression models can predict valence and arousal levels based on raw sensor data and how the normalization of signals and target values affects the recognition rates. From these results, it can be noted that it is possible to reliably estimate valence and arousal levels based on data from wrist-worn wearable sensors and up-to-date prediction models. Moreover, this estimation is especially reliable when the prediction is made based on the LSTM model. It should be noted that the LSTM outperforms other classification and regression algorithms. The best results were obtained using the LSTM regression model with a baseline reduction as the normalization method, in this case, for valence level estimation MSE = 0.43 and $R^{2}$ = 0.71 and for arousal level estimation MSE = 0.59 and $R^{2}$ = 0.81.

A comparison of the results from the classification and regression models shows that, in general, the regression models performed better than the classification models, and only in very few cases did classification models perform better than regression models. This is not surprising, as valence and arousal are continuous phenomena and are not discrete, meaning that they should be analyzed using regression methods, not classification methods. However, in certain cases classification using LSTM worked very well. For instance, when the valence level is recognized, the LSTM-based classification model with baseline reduction normalization (mean MSE = 0.57 and $R^{2}$ = 0.55) performs nearly as well as the LSTM-based regression model with baseline reduction normalization, which has the overall best MSE score (0.43). In addition, when the arousal level is predicted using the LSTM-based classification model with baseline reduction normalization, the performance of the model is nearly as good as when using the LSTM-based regression model with baseline reduction normalization (MSE = 0.81 and $R^{2}$ = 0.75 compared to MSE = 0.59 and $R^{2}$ = 0.81). Therefore, it is not possible to conclude based on MSE and $R^{2}$ that LSTM-based regression models are better than LSTM-based classification models. To study the performance of the LSTM-based models in more detail and compare their classification and regression versions, Table 4 presents a comparison using MSE and $R^{2}$ along with RMSE and MAE. According to these results, baseline reduction is the best normalization method, supporting the findings based on the results of Table 3. Moreover, according to Table 4, in the case of valence recognition the difference between LSTM-based classification and regression models with baseline reduction is small when MSE, $R^{2}$, RMSE, and MAE values are compared. Nonetheless, when all four performance metrics are compared, the LSTM regression model with baseline reduction is better than the most similar classification model according to three metrics out of four. In the case of arousal recognition, the difference is clear, and again the LSTM based regression model with baseline reduction is the best model according to three metrics out of four.

According to Table 4, the two best models are LSTM regression and classification models with baseline reduction. To obtain more insight into these models, Figure 5 and Figure 6 illustrate how the predicted valance and arousal estimates follow the user-reported target variables when these models are used in prediction. The figures are drawn based on the results of the best runs; in the case of the regression model, MSE and $R^{2}$ for valence estimation were 0.38 and 0.74, respectively, while for arousal estimation they were 0.51 and 0.84, respectively. For the classification model, the MSE and $R^{2}$ for valence estimation were 0.42 and 0.69, respectively, while for arousal estimation they were 0.68 and 0.80, respectively. In the figures, predictions using an LSTM-based regression model are shown with a blue line, those using a classification model are shown using a green line, and the true arousal level is shown in orange. Due to subjective differences, the estimation is not as good for all subjects; however, these figures show that in general prediction is highly accurate with both models. In fact, for a number of study subjects the prediction is almost perfect. However, while the difference between LSTM regression and classification models according to the MSE and $R^{2}$ is minimal, Figure 5 and Figure 6 reveal differences. It can be noted that the WESAD data does not contain very many samples from cases in which the level of valence is very high or very low, and it contains very few negative arousal cases. In fact, Figure 5 shows that the models have difficulty detecting high valence values; in particular, the classification models seem to suffer due to this lack of training data for high valence values. According to Figure 5, the classification model performs badly for samples in which valence is above zero, while the regression model has fewer such problems. Similarly, Figure 6 shows that the classification model has problems detecting high arousal values; here, the problems are not as severe as in the case of valence recognition, as the training data contain more cases with high values for arousal than for valence. In addition, according to Figure 6, neither model detects negative arousal samples.

Earlier results have already shown that baseline reduction is the most effective normalization method. However, when different normalization methods are compared, it is especially interesting to see the effects different normalization methods have on LSTM models, as these outperform other models. This is visualized in Figure 7. The results of this figure are taken from Table 4 by calculating the average performance of each normalization method when LSTM classification and regression models are used to detect valence and arousal levels. The figure clearly shows that there are large differences between the normalization methods; no matter which performance metric is used, baseline reduction always provides the best results. For MSE, RMSE, and MSE the error is the lowest and for $R^{2}$ the value is highest when using baseline reduction. In fact, according to Table 4, for the cases of both valance, and arousal the best results are obtained when using baseline reduction as the normalization method. Both classification and regression models benefit from this, showing that normalization should be used instead of analyzing raw data. The low performance of *z*-score normalization is surprising; it provides good results only rarely, and in this study, the only good results using *z*-score normalization were obtained when the valence level was detected using an LSTM-based regression model (MSE = 0.70 and $R^{2}$ = 0.75, see Table 4). While *z*-score normalization does not perform well compared to baseline reduction, it is a much better option than analyzing data without any normalization. In fact, Figure 7 shows that, on average, the worst results were obtained from raw data, with the performance of non-normalized data being especially bad according to the RMSE value.

Figure 5 and Figure 6 show that the valence and arousal levels can be estimated with high reliability when studied separately, and an LSTM-based regression model with baseline reduction is the best method to do it. However, the most important thing is to understand how well emotions can be estimated when valence and arousal estimations are combined and visualized using Russell’s circumplex model of emotions (see Figure 1). Figure 8 shows this visualization for different emotion classes; these estimations are from the run that provided the best results when the target values were normalized using baseline reduction. Therefore, they are the same ones shown in Figure 5 and Figure 6 for the LSTM-based regression model. In Figure 8, the estimated values are shown in blue and the target values provided by the study subjects are visualized using red dots. As baseline reduction is used, in the case of the baseline class the target value for valence and arousal is zero. Figure 8a shows that the baseline emotion can be estimated with high accuracy, as almost all the estimations are close to the origin. In this case, the average estimated valence is −0.01 and the average estimated arousal is 0.04. According to Figure 1, strong negative emotions are located at the top left quarter of Russell’s circumplex model of emotions, which is exactly where estimations of stress-class observations are located based on the models presented in this article (see Figure 8b). Moreover, the target values obtained from the study subjects are located in the same place. In fact, the predicted values and target values are very close to each other. Observations from the amusement class are estimated to be located close to the origin (Figure 8c) or to the right bottom quarter of Russell’s circumplex model of emotions, where relaxed emotions are located. While the model estimates only slightly relaxed emotions during the amusement class, and the detected emotions are not as strong as those recognized from the stress class, this does not mean that the model performs badly in this case. Indeed, when predicted values are compared to the target values, it can again be noted that they are distributed in the same area on the valence–arousal graph. Therefore, prediction models based on the LSTM regression model and baseline reduction can estimate the valence and arousal levels for each emotion class with high accuracy, making it emotion-independent based on this analysis.

### 5.2. Subject-Wise Results

Subject-wise valence and arousal level estimation results from the best-performing regression models are presented in Table 5, where LSTM models without any normalization and with baseline reduction are compared to AdaBoost and Random Forest models with baseline reduction. It should be noted that, according to Table 3, the AdaBoost and Random Forest models perform much worse on average than the LSTM models. The results in Table 5 show that for most of the study subjects the levels of valence and arousal can be predicted by all of these models, as well as with the AdaBoost and Random Forest models. There are even cases in which AdaBoost and Random Forest perform better than LSTM. However, the largest difference between AdaBoost, Random Forest, and LSTM is that in certain cases AdaBoost and Random Forest perform very badly, while the variance between the prediction rates for different study subjects is much smaller using LSTM. For instance, when the valence of subject 11 was predicted using the AdaBoost regression model, the $R^{2}$-score was −131.95, and for Random Forest the $R^{2}$-score was −127.81. These naturally have a huge effect on the average values presented in Table 3.

The results in Table 5 show that certain study subjects have data that are more difficult to predict. For instance, each model has difficulty predicting the valence of study subjects 14 and 17 and the arousal of study subjects 2, 14, and 17. There may be problems with the data of study subjects 14 and 17, or their bodies may react differently to stimuli compared to other study subjects. If the differences are caused by different stimuli, this suggests that it would be possible to obtain better results via model personalization. In addition, there are model-specific differences. For instance, the valence level of study subject 4 is not predicted well by LSTM when the model uses raw data; however, when the same person’s data is predicted with the LSTM model trained using baseline reduction normalized data, the prediction is highly accurate. This shows the importance of normalization. Moreover, while LSTM performs well in most cases, for certain subjects the $R^{2}$-score is negative.

For this experiment, all of the models were trained five times; the results presented in this section are averages from these runs, with the standard deviation from these runs for each individual presented in parentheses in Table 5. When the standard deviations are studied in detail, it can be noted that for certain study subjects the results differ a great deal between different runs, especially when it comes to valence level detection. For instance, for study subjects 2, 5, and 14 the standard deviation of the $R^{2}$ score is greater than 1 when valence is detected using LSTM and baseline reduction.

### 5.3. Experimenting with Sensor Combinations

Different sensor combinations were compared to study the effects of different sensors on the recognition results. The results calculated using the LSTM regression model with baseline reduction are shown in Table 6. Table 6 shows that not all of the Empatica E4’s sensors are needed to estimate valence levels reliably, and arousal can be estimated at a high rate without using all the sensors as well. In fact, when using just the BVP and EDA sensors the valence levels can be estimated with the same detection rate as when using all the sensors. When these results are studied subject-wise, it can be noted that the variance between the study subjects is smaller when using only BVP and EDA sensors instead of all the sensors (see Table 7). When the LSTM regression model with baseline reduction ws used with all the sensors to recognize valence level, the $R^{2}$ score was negative for four study subjects (see Table 5). However, according to Table 7, the $R^{2}$ score is negative only for one study subject when using only the BVP and EDA sensors. In addition, the variance within the study subjects is smaller when using just the BVP and EDA sensors instead of all the sensors; for instance, the variance of the $R^{2}$-score varies from 0 to 0.14 depending on the study subject when using only the BVP and EDA sensors, while when using all the sensors it varies from 0.01 to 1.10.

The recognition rate of the arousal level is slightly lower when using just EDA and ST instead of all the sensors (Table 6). In this case, the selection of only certain sensors does not have a similar positive effect on standard deviation as does for valence; again, however, the results in this case are better for certain individuals.

## 6. Discussion

The results presented in Section 5 show that fine-grained valence and arousal levels can be estimated reliably based on wrist-worn wearable sensor data and machine learning methods. According to the results shown in Table 3, LSTM models are superior to other methods. This is because LSTM is the only prediction model in our experiments capable of learning long-term dependencies from the data. Moreover, LSTM is the most advanced among these prediction models, which is why its superior performance on the WESAD dataset is not surprising. As shown in Figure 2, the WESAD dataset is complex, and it requires powerful methods for analysis. Especially in the case of estimating the arousal level, the difference between LSTM and other prediction methods is very large.

Table 3 shows that regression models perform better for the task in general than classification models when the performance of the models is measured using MSE and $R^{2}$ values. This is expected, as valence and arousal are continuous phenomena and are not discrete. Moreover, as classification methods treat valence and arousal levels as distinct categories, they are unable to take into account the ordinal nature of these levels and use it during the model training process. This means that if the training data do not contain samples from all the possible levels of valence and arousal, as in the case of our data, classification models cannot detect these from unseen datasets as well. Regression models do not have this limitation. Despite their limitations, in certain cases classification methods performed well in our experiments. In fact, according to MSE and $R^{2}$ values, the LSTM-based classification model with baseline reduction performs equally as well as the regression-based LSTM model, though the RMSE and MAE measures (Table 4) and visualization of the results (Figure 5 and Figure 6) show that there are in fact differences between these models and that the LSTM regression model is more reliable than the LSTM classification model. The biggest difference is that the LSTM-based classification methods seem to have problems detecting positive valence values. In fact, the dataset does not contain many such samples, showing that the LSTM classification model is more vulnerable to limited and imbalanced datasets than the LSTM regression model. Moreover, when the predicted values were visualized using Russell’s circumplex together with target values provided by the study subjects (see Figure 8), it can be noted that the results are almost identical. This shows that the models presented in this article are highly capable of recognizing the level of valence and arousal and that the valence and arousal level estimates provided by the study subjects contain information that can be used as target values for the recognition models when data are pre-processed and normalized correctly. However, it seems that the targets reported by the study subjects are not always reliable. For instance, subject 3 reported a valence label of 7 after a stress condition, which was the same as this person reported for baseline valence, claiming that he was looking forward to the next condition and was therefore cheerful. The models did not manage to predict this correctly; the results using LSTM, AdaBoost, and Random Forest are as shown in Figure 9. However, as the target value defined by the study subject was not reliable, most likely the estimations made by the models are closer to the truth than the target variables. Moreover, it is possible that people might not always know how they feel [43,44], and for this reason it seems that in certain cases the models are better at describing feelings than the study subjects themselves.

In addition, different methods for data normalization were used: *z*-score, baseline reduction, and *z*-score with baseline reduction. These were compared to the case where features were extracted from raw (non-normalized) data. The results were surprising, in that the best results were obtained using baseline reduction normalization. It was expected that *z*-score normalization would provide the best results, as it has been shown to improve the detection rates when stress or other affective states are recognized from wearable sensor data [45]. However, in this study no similar effect was noted. The reason for this could be that the previous studies concentrated on detecting discrete human emotions and not on estimating continuous valence and arousal values, as this study does. A baseline reduction-based *z*-score normalization has other advantages over *z*-score normalization as well; in order to normalize signals using *z*-score normalization, the participant-specific mean and standard deviation need to be calculated from each study subject’s full data signal, meaning that normalization can only be carried out after data gathering [46]. Due to this, it is not suitable for real-time application. However, baseline reduction does not have a similar limitation, as only baseline data need to be collected, and the study subject can report his/her valence and arousal levels at the same time. Moreover, as baseline reduction improves recognition rates and *z*-score normalization does not, this means that the differences between the individuals are more related to differences in subjective target variable estimations than to differences in the signals themselves.

Our in-depth subject-wise study (see Table 5) shows that for most of the study subjects the AdaBoost and Random Forest models perform almost as well as LSTM. However, the biggest difference between these models is that for certain subjects the AdaBoost and Random Forest models perform very badly, while the LSTM-based model is more evenly reliable for each individual. Therefore, LSTM does not suffer from much variance between the study subjects, resulting in much better average prediction rates than all the other experimented models. These results show that machine learning is a powerful tool for detecting valence and arousal levels, and thereby for recognizing emotions. While the results using the LSTM regression model are good on average, even in the case of the LSTM regression model there is variation between the study subjects and different runs. This means that there is a need to personalize models and experiment with larger datasets that have more variation. Moreover, the results presented in Table 7 show that when using only certain sensors it is possible to obtain estimations that are just as good as when using all the sensors. For certain individuals, the results are even better, especially when it comes to detecting the valence level, as in this case the variance between the study subjects can be reduced when using only the EDA and BVP sensors instead of all the sensors. Therefore, prediction models could be personalized by selecting a unique sensor combination for each individual, which could improve the results and reduce the variance between study subjects.

## 7. Conclusions

This study aimed to predict emotions using biosignals collected via wrist-worn sensor and to evaluate the accuracy of different prediction models. Two dimensions of emotions were considered, namely, valence and arousal. In this study, valence and arousal levels were estimated using machine learning methods based on an open-access WESAD dataset containing biosensor data from wrist-worn sensors. These data included skin temperature, electrodermal activity, blood volume pulse, heart rate, and heart rate variability collected from 15 study subjects. Study subjects were exposed to different stimuli (baseline, stress, and amusement); after each stimulus, they reported their valence and arousal levels. These estimates were used in this study as target variables. In fact, in the study it was shown that the level of valence and arousal can be predicted with high reliability with the help of these user-reported valence and arousal levels by using the LSTM regression model and normalizing target values through baseline reduction. However, while on average the results are very good, for certain individuals the results are much weaker. Moreover, it was found that reliable models could be obtained even if all biosignals were not used in the training phase; in fact, for certain study subjects the best results were obtained using only a few of the sensors.

To date, the field of emotion detection has mainly focused on identifying a limited number of discrete emotions or treating valence and arousal as coarse-grained discrete variables. However, this study demonstrates the ability to reliably detect fine-grained valence and arousal levels by analyzing data using advanced machine learning models. Additionally, this research suggests that regression models are more effective for this task than classification models. By recognizing emotions through the dimensions of valence and arousal rather than discrete emotions, this study takes a step towards a more sophisticated and nuanced understanding of emotion detection using biosignals and machine learning. This shift towards analyzing continuous variables rather than discrete emotions is expected to be the focus of future research in this field.

However, this study has weaknesses as well. When subject-wise results were studied, it was noted that there was variance between the recognition rates for the different individuals; therefore, the recognition rates were not equally good for each study subject. In certain cases, this variance between individuals could be reduced by personalizing the models by selecting a unique sensor combination for each individual. In fact, one future tasks is to study model personalization in more detail; for instance, incremental learning could be an effective method to personalize models based on streaming data [47]. Moreover, feature selection needs to be studied in order to provide reliable estimates for each individual. For instance, a sequential backward floating search has been found to be an effective feature selection method for biosignals [34].

Another weakness of this study is that the experiments were based on only one dataset; due to this, future work needs to include experimenting with other datasets. In particular, it should be studied how well negative arousal levels can be estimated, as the WESAD dataset used in this study contained very few negative arousal values. Moreover, the LSTM model used in this article was quite simple; it had one hidden layer, with 64 units and around 17,000 trainable parameters. However, if the aim is to build a model that can be run in real-time in a wrist-work device, a complexity analysis should be carried out in order to better understand how much calculation capacity this type of model requires. In addition, the parameters of the prediction models should be tuned in order to optimize the results, as in this study the prediction models were trained without parameter tuning.

## Figures and Tables

**Figure 1 sensors-23-01598-f001:**
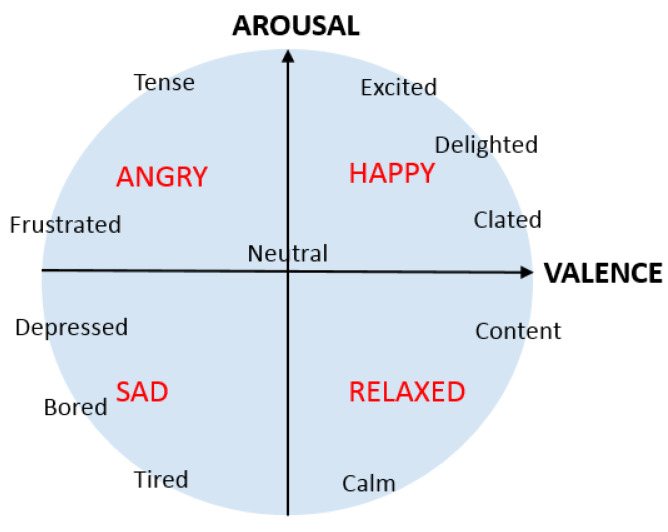
Russell’s Circumplex Model of Emotions.

**Figure 2 sensors-23-01598-f002:**
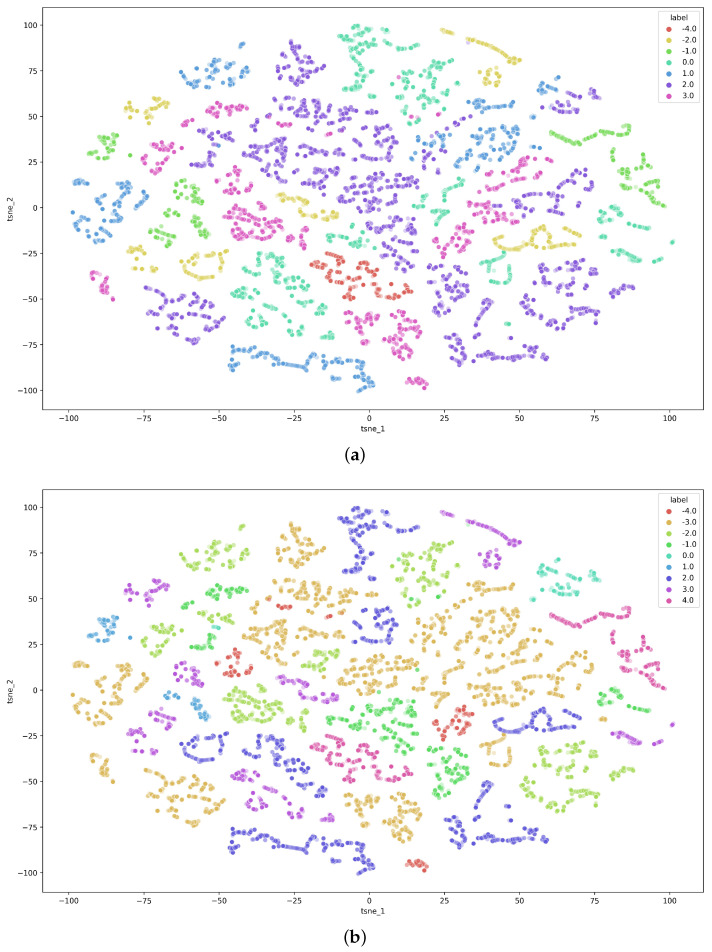
WESAD dataset illustrated using t-SNE: (**a**) data visualized using t-SNE and valence levels as targets; (**b**) data visualized using t-SNE and arousal levels as targets.

**Figure 3 sensors-23-01598-f003:**
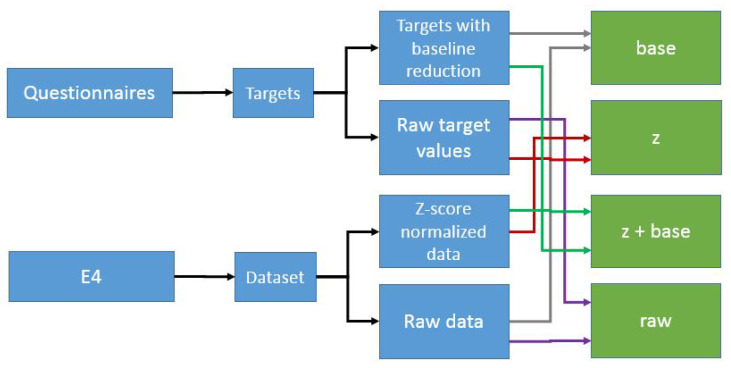
Four datasets (base, z, z+base, and raw) were created to experiment with different normalization methods.

**Figure 4 sensors-23-01598-f004:**
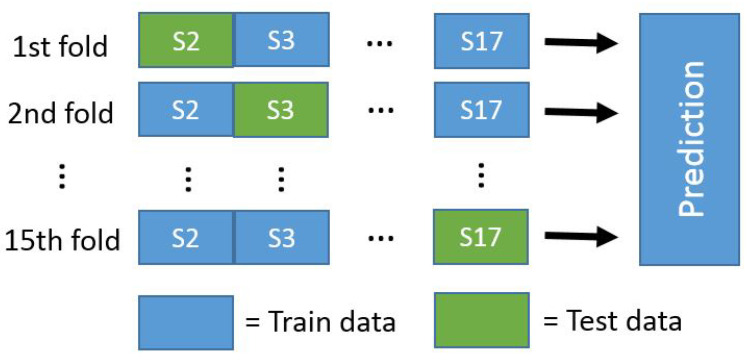
Leave-one-subject-out method used in the experiments. In turns, the data of one study subject are is used for testing while all other data are used for [42].

**Figure 5 sensors-23-01598-f005:**
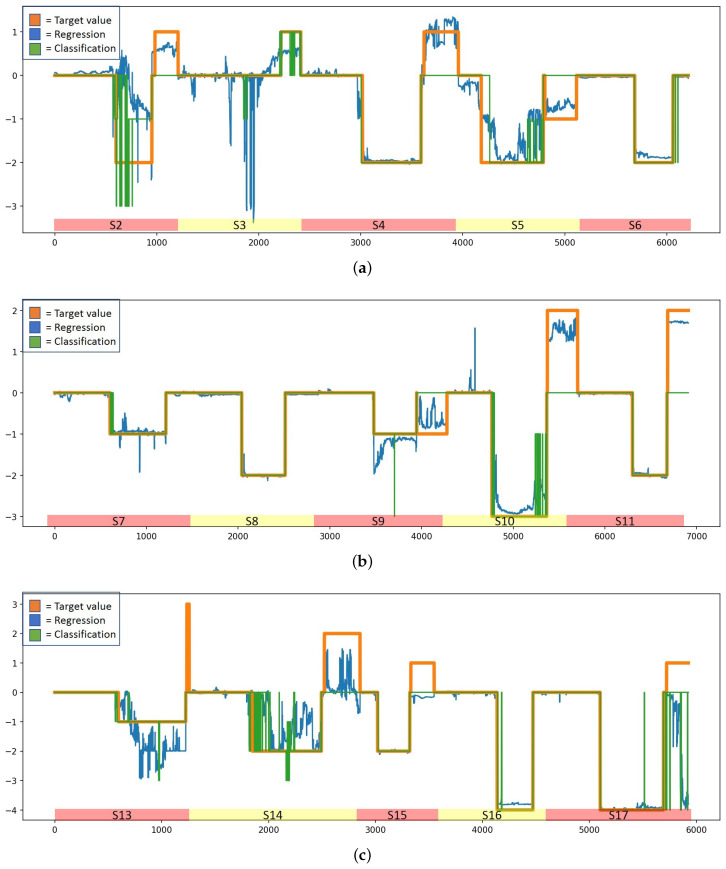
True and predicted valence levels for the WESAD dataset using LSTM-based regression (blue line) and classification (green line) models with baseline normalization. The true valence level is shown in orange. (**a**) Valence for study subjects 2, 3, 4, 5, and 6. (**b**) Valence for study subjects 7, 8, 9, 10, and 11. (**c**) Valence for study subjects 13, 14, 15, 16, and 17.

**Figure 6 sensors-23-01598-f006:**
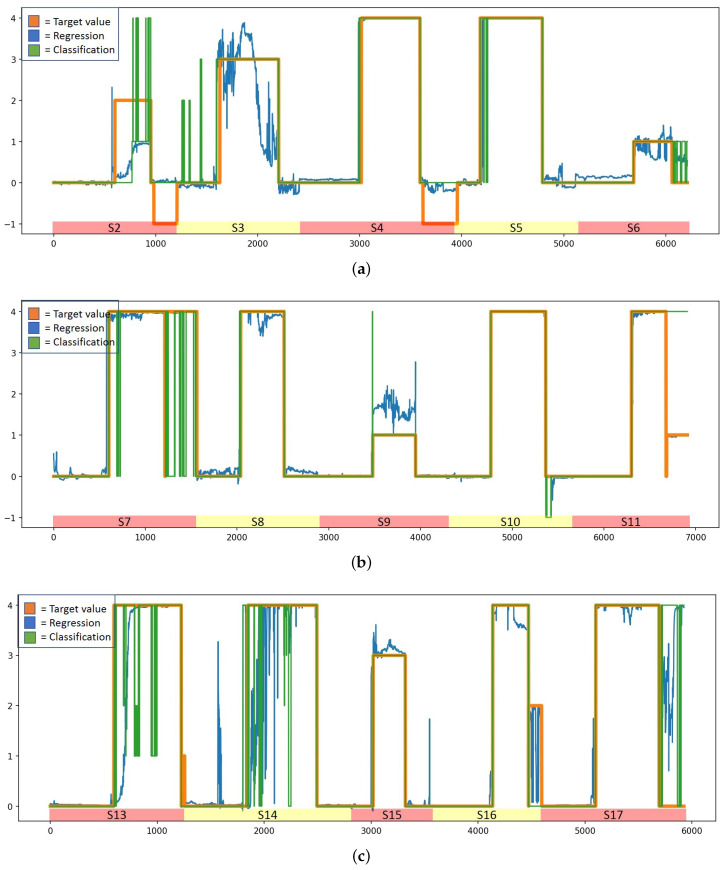
True and predicted arousal levels for the WESAD dataset using LSTM-based regression (blue line) and classification (green line) models with baseline normalization. The true arousal level is shown in orange color. (**a**) Arousal for study subjects 2, 3, 4, 5, and 6. (**b**) Arousal for study subjects 7, 8, 9, 10, and 11. (**c**) Arousal for study subjects 13, 14, 15, 16, and 17.

**Figure 7 sensors-23-01598-f007:**
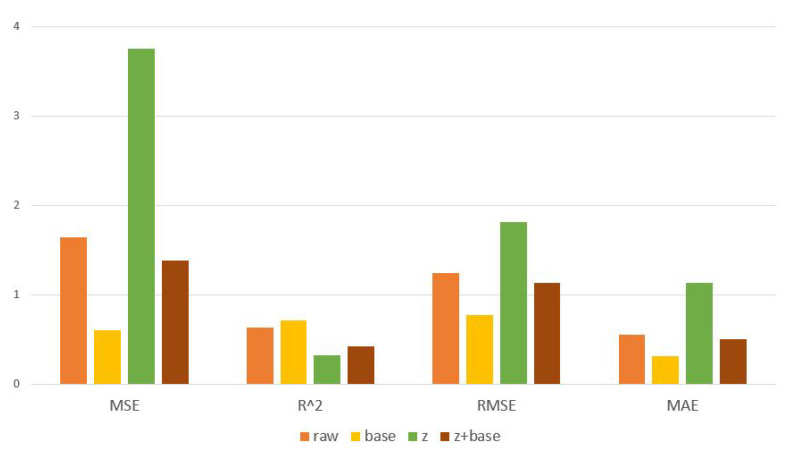
Effect of different normalization methods on the performance of the LSTM models.

**Figure 8 sensors-23-01598-f008:**
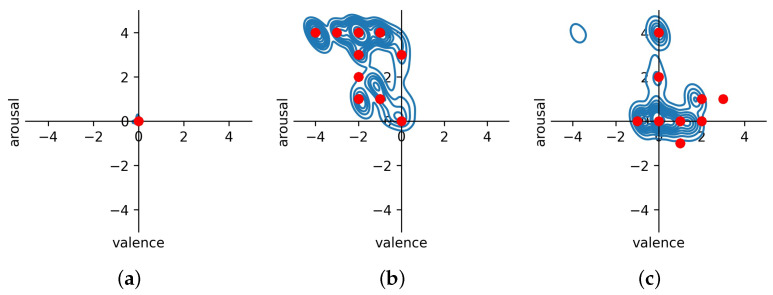
Results from different emotion classes. The results are presented on a valence–arousal graph, with valence on the *x*-axis and arousal on the *y*-axis. Study subjects’ reported valence and arousal levels are shown as red dots, while the blue graphs indicate the distribution of estimated values across the graph: (**a**) baseline; (**b**) stress; (**c**) amusement.

**Figure 9 sensors-23-01598-f009:**
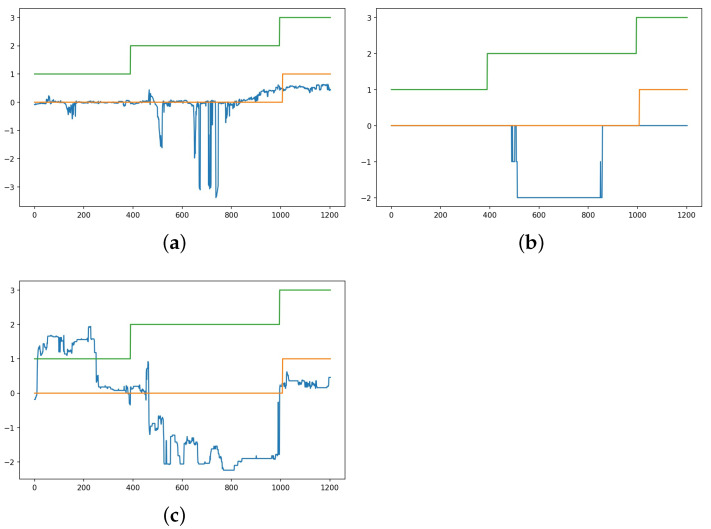
Subject 3 might have provided the wrong answer regarding the level of valence during a stressful stage. The blue line shows the predicted valence level, the orange line is the ground truth for the valence level, and the green line indicates the emotion class: 1 = baseline, 2 = stress, and 3: amusement. (**a**) LSTM, (**b**) AdaBoost, (**c**) Random Forest.

**Table 1 sensors-23-01598-t001:** Previous research utilizing wearable sensor data to identify valence, arousal, and other related affect states.

Article	Signals	What Is Recognized	Continuous/Discrete Targets	Methods
Hsu et al. (2017) [11]	ECG	valence and arousal	discrete (high/low)	least squares SVM
Romeo et al. (2019) [12]	2 settings: 1. EEG + 13 other biosignal, 2. GSR, IBI and ST	valence and arousal	discrete (high/low)	Multiple Instance Learning, SVM, Random Forest, Naive Bayes
Li et al. (2020) [13]	EEG, EOG, EMG, EDA, BVP, ST, and respiration	valence and arousal	discrete (high/low)	DNN
Zhao et al. [14]	BVP, EDA, ST	valence and arousal	discrete (high/low)	SVM
Diaz-Romero et al. (2021) [15]	EEG, EOG, motion	valence and arousal	discrete (2 and 3 class cases)	SVM, logistic regression, Random Forest, and ensemble classifier
Zhang et al. (2020) [16]	2 settings: 1. ECG, BVP, EDA, respiratory rate, ST, and EMG, 2. HR, HRV, ST, and EDA + data from wearable eyetracker	valence and arousal	discrete (2, 3 and 4 class cases)	correlation-based emotion recognition algorithm (CorrNet)
Hwang et al. (2018) [17]	EEG	valence	continuous	statistical methods, correlation
Nirjhar et al. (2020) [19]	BVP, ST, EDA, audio	anxiety	continuous	statistical methods such as least square regression
Grundlehner et al. (2009) [18]	ECG, respiration rate, EDA and ST	arousal	continuous	linear regression
Siirtola & Röning (2020) [20]	BVP, EDA, ST	stress	continuous	Bagged tree based ensemble for regression
Healey & Picard (2005) [21]	ECG, EMG, EDA, and respiration rate	stress	continuous	statistical methods such as correlation
El Haouij (2018) [22]	EDA from both wrists, HR, respiratory rate, posture	stress	continuous	Random Forest for regression
Vos et al. (2022) [23]	BVP, HR, EDA, ST	stress	continuous	(XGBoost) XGB, (artificial neural networks) ANN, XGB+ANN

**Table 2 sensors-23-01598-t002:** List of extracted features.

Signal	Features Extracted from the Pre-Processed Biosignals
Electrodermal activity (EDA)	mean, std, min, max, median, range, percentile_95, percentile_5, percentile_75, percentile_25, phasic_mean, phasic_std, phasic_min, phasic_max, phasic_median, phasic_range, phasic_percentile_95, phasic_percentile_5, phasic_percentile_75, phasic_percentile_25, tonic_mean, tonic_std, tonic_min, tonic_max, tonic_median, tonic_range, tonic_percentile_95, tonic_percentile_5, tonic_percentile_75, tonic_percentile_25,
Skin temperature (ST)	mean, std, min, max, percentile_95, percentile_5, percentile_75, percentile_25, slope
Blood volume pulse (BVP)	mean, std, min, max, percentile_95, percentile_5, percentile_75, percentile_25 HR_mean, NNi_mean, NNi_std, NNi_length, NNi50, pNN50, TINN, rms, LF/HF_ratio, VLF_relative_power, LF_relative_power, HF_relative_power, VHF_relative_power, total_power, UHF_energy, LF_energy, HF_energy, breathing_rate

**Table 3 sensors-23-01598-t003:** Comparison of classification (C) and regression (R) models to predict valence and arousal level based on MSE and $R^{2}$ scores (standard deviation in parentheses). Rows show results using different prediction models and columns show results using different normalization methods (raw = no normalization, z = *z*-score normalization, base = baseline reduction, z + base = *z*-score normalization and baseline reduction).

**User-Independent Prediction Rates for Valence Level Estimation.**					
**Method**	**Type**	**Raw: MSE/** $R^{2}$	**z: MSE/** $R^{2}$	**Base: MSE/** $R^{2}$	**z + Base: MSE/** $R^{2}$
LSTM	C	1.75 (0.53)/0.42 (0.18)	3.33 (0.16)/0.13 (0.07)	0.57 (0.11)/0.55 (0.10)	1.94 (0.18)/−0.04 (0.07)
LDA	C	4.88 (0.00)/0.39 (0.00)	2.96 (0.04)/−0.20 (0.02)	4.16 (0.0)/−0.20 (0.0)	2.63 (0.00)/−0.68 (0.00)
Decision Tree	C	4.04 (0.41)/−0.26 (0.09)	4.46 (0.12)/−0.32 (0.03)	2.54 (0.13)/−0.45 (0.04)	3.23 (0.10)/−1.07 (0.10)
AdaBoost	C	3.78 (0.04)/−0.54 (0.05)	3.83 (0.06)/−0.52 (0.04)	1.63 (0.02)/−0.37 (0.09)	3.21 (0.09)/−1.30 (0.05)
Random Forest	C	4.48 (0.14)/−0.75 (0.06)	3.41 (0.04)/−0.46 (0.07)	1.85 (0.04)/−0.63 (0.11)	2.59 (0.12)/−1.34 (0.07)
XGBoost	C	3.79 (0.10)/−1.15 (0.06)	3.02 (0.03)/−0.44 (0.02)	1.79 (0.05)/−0.61 (0.04)	2.02 (0.10)/−0.58 (0.01)
LSTM	R	0.74 (0.20)/ 0.73 (0.08)	0.70 (0.09)/**0.75 (0.04)**	**0.43 (0.05)**/0.71 (0.04)	0.57 (0.06)/0.61 (0.05)
AdaBoost	R	4.51 (0.19)/−0.79 (0.14)	3.41 (0.08)/−0.40 (0.00)	1.54 (0.04)/−1.07 (0.08)	2.05 (0.06)/−0.80 (0.05)
Histogram-based GBM	R	2.86 (0.05)/−0.91 (0.08)	2.79 (0.14)/−0.36 (0.04)	1.74 (0.03)/−0.55 (0.02)	2.03 (0.03)/−0.59 (0.01)
Linear model	R	3.28 (0.00)/0.10 (0.00)	2.20 (0.00)/−0.26 (0.00)	4.60 (0.0)/−0.27 (0.0)	1.47 (0.0)/−0.37 (0.0)
Random Forest	R	3.43 (0.06)/−1.14 (0.05)	3.06 (0.04)/−0.38 (0.03)	1.85 (0.04)/−0.62 (0.03)	2.08 (0.02)/−0.59 (0.01)
XGBoost	R	3.57 (0.11)/−1.65 (0.08)	2.86 (0.06)/−0.51 (0.03)	1.70 (0.02)/−0.87 (0.03)	2.01 (0.04)/−0.68 (0.06)
**User-Independent Prediction Rates for Arousal Level Estimation.**					
**Method**	**Type**	**Raw: MSE/** $R^{2}$	**z: MSE/** $R^{2}$	**Base: MSE/** $R^{2}$	**z + Base: MSE/** $R^{2}$
LSTM	C	2.51 (0.79)/0.64 (0.11)	8.41 (0.44)/−0.17 (0.07)	0.81 (0.10)/0.75 (0.03)	1.75 (0.19)/0.50 (0.05)
LDA	C	9.48 (0.00)/−0.56 (0.00)	4.86 (0.00)/0.24 (0.00)	3.08 (0.0)/0.00 (0.0)	2.37 (0.0)/0.29 (0.0)
Decision Tree	C	6.72 (0.54)/−0.27 (0.04)	7.08 (0.04)/−0.21 (0.01)	3.16 (0.16)/−0.20 (0.05)	2.81 (0.08)/0.02 (0.02)
Random Forest	C	6.22 (0.21)/−0.25 (0.07)	7.38 (0.04)/−0.21 (0.01)	3.08 (0.09)/−0.125 (0.02)	3.02 (0.07)/0.01 (0.02)
AdaBoost	C	6.05 (0.20)/−0.19 (0.03)	6.57 (0.22)/−0.06 (0.02)	2.81 (0.10)/0.11 (0.05)	2.56 (0.26)/0.23 (0.07)
XGBoost	C	6.31 (0.23)/−0.18 (0.06)	6.79 (0.07)/−0.11 (0.01)	3.37 (0.12)/−0.10 (0.025)	2.75 (1.56)/0.36 (0.20)
LSTM	R	1.57 (0.14)/0.73 (0.03)	2.56 (0.36)/0.58 (0.07)	**0.59 (0.06)**/**0.81 (0.02)**	1.24 (0.24)/0.59 (0.07)
AdaBoost	R	6.01 (0.25)/−0.16 (0.03)	6.32 (0.34)/−0.04 (0.05)	2.63 (0.16)/0.18 (0.05)	2.78 (0.05)/0.16 (0.03)
Histogram-based GBM	R	5.98 (0.16)/−0.18 (0.06)	4.96 (0.15)/−0.14 (0.03)	2.48 (0.09)/−0.06 (0.02)	2.55 (0.07)/−0.08 (0.02)
Linear model	R	7.18 (0.00)/0.03 (0.00)	3.53 (0.00)/0.19 (0.00)	2.37 (0.0)/0.09 (0.0)	1.71 (0.0)/0.20 (0.0)
Random Forest	R	5.90 (0.09)/−0.48 (0.02)	4.79 (0.10)/−0.18 (0.02)	2.18 (0.06)/0.02 (0.03)	2.51 (0.01)/−0.17 (0.01)
XGBoost	R	5.43 (0.25)/−0.26 (0.05)	4.75 (0.06)/−0.16 (0.02)	2.23 (0.07)/−0.09 (0.04)	2.58 (0.02)/0.19 (0.01)

**Table 4 sensors-23-01598-t004:** Detailed analysis of LSTM classification (C) and regression (R) models with different normalization methods.

**User-Independent Prediction Rates for Valence Level Estimation.**					
**Method**	**Type**	**MSE**	$R^{2}$	**RMSE**	**MAE**
LSTM raw	R	0.74 (0.20)	0.73 (0.08)	0.82 (0.12)	0.42 (0.08)
LSTM base	R	**0.43 (0.05)**	0.71 (0.04)	**0.65 (0.04)**	0.34 (0.05)
LSTM z	R	0.70 (0.09)	**0.75 (0.04)**	0.83 (0.06)	0.41 (0.06)
LSTM z+base	R	0.57 (0.06)	0.61 (0.05)	0.76 (0.04)	0.40 (0.04)
LSTM raw	C	1.75 (0.53)	0.42 (0.18)	1.42 (0.16)	0.63 (0.12)
LSTM base	C	0.57 (0.11)	0.55 (0.10)	0.75 (0.08)	**0.30 (0.04)**
LSTM z	C	3.33 (0.16)	0.13 (0.07)	1.90 (0.07)	1.27 (0.40)
LSTM z+base	C	1.94 (0.18)	−0.04 (0.07)	1.34 (0.07)	0.55 (0.05)
**User-Independent Prediction Rates for Arousal Level Estimation.**					
**Method**	**Type**	**MSE**	$R^{2}$	**RMSE**	**MAE**
LSTM raw	R	1.57 (0.14)	0.73 (0.03)	1.25 (0.06)	0.60 (0.08)
LSTM base	R	**0.59 (0.06)**	**0.81 (0.02)**	**0.77 (0.04)**	0.30 (0.03)
LSTM z	R	2.56 (0.36)	0.58 (0.07)	1.60 (0.11)	0.95 (0.10)
LSTM z+base	R	1.24 (0.24)	0.59 (0.07)	1.11 (0.11)	0.51 (0.06)
LSTM raw	C	2.51 (0.79)	0.64 (0.11)	1.48 (0.11)	0.55 (0.07)
LSTM base	C	0.81 (0.10)	0.75 (0.03)	0.90 (0.05)	**0.28 (0.03)**
LSTM z	C	8.41 (0.44)	−0.17 (0.07)	2.90 (0.08)	1.90 (0.12)
LSTM z+base	C	1.75 (0.19)	0.50 (0.05)	1.32 (0.07)	0.55 (0.06)

**Table 5 sensors-23-01598-t005:** Subject-wise prediction rates (standard deviation in parentheses) for valence and arousal detection using different prediction models and normalization methods.

**Subject-Wise Prediction Rates (Standard Deviation in Parentheses) for Valence Detection.**				
**Subject**	**LSTM Raw: MSE/** $R^{2}$	**LSTM Base: MSE/** $R^{2}$	**AdaBoost Base: MSE/** $R^{2}$	**Random Forest Base: MSE/** $R^{2}$
2	1.50 (0.20)/−2.92 (2.26)	0.83 (0.25)/−1.31 (1.10)	0.71 (0.01)/−0.37 (0.05)	0.51 (0.20)/0.07 (0.45)
3	0.51 (0.25)/−0.02 (0.16)	0.50 (0.15)/−0.26 (0.18)	1.48 (0.12)/−0.65 (0.09)	1.62 (0.26)/−0.37 (0.20)
4	0.80 (0.19)/0.57 (0.12)	0.22 (0.27)/0.79 (0.28)	0.42 (0.08)/0.75 (0.04)	0.78 (0.28)/0.74 (0.03)
5	0.06 (0.02)/0.88 (0.02)	0.41 (0.17)/−1.52 (2.27)	0.94 (0.07)/−2.07 (0.76)	0.96 (0.08)/−1.50 (0.91)
6	0.05 (0.03)/0.92 (0.05)	0.02 (0.01)/0.97 (0.01)	0.18 (0.09)/0.59 (0.25)	0.10 (0.09)/0.78 (0.26)
7	0.10 (0.10)/0.83 (0.11)	0.03 (0.03)/0.88 (0.08)	0.35 (0.09)/0.56 (0.06)	0.29 (0.11)/0.57 (0.04)
8	0.00 (0.00/1.00 (0.00)	0.01 (0.00)/0.99 (0.00)	0.31 (0.19)/0.21 (0.79)	0.35 (0.17)/0.08 (0.71)
9	0.04 (0.04)/0.82 (0.17)	0.19 (0.09)/0.60 (0.11)	0.62 (0.04)/0.32 (0.02)	0.79 (0.17)/0.31 (0.02)
10	0.52 (0.53)/0.90 (0.10)	0.20 (0.08)/0.93 (0.04)	2.77 (0.17)/−5.19 (2.12)	2.91 (0.19)/−8.41 (2.75)
11	0.22 (0.14)/0.85 (0.10)	0.04 (0.01)/0.98 (0.01)	1.85 (0.08)/−131.95 (201.36)	2.10 (0.28)/−127.81 (203.78)
13	0.48 (0.13)/0.60 (0.05)	0.78 (0.13)/0.31 (0.05)	0.67 (0.03)/0.24 (0.04)	0.49 (0.18)/0.23 (0.05)
14	2.87 (1.48)/−1.90 (1.89)	1.28 (0.19)/−1.56 (1.12)	2.27 (0.14)/−26.71 (20.60)	2.06 (0.28)/−13.33 (9.17)
15	0.05 (0.01)/0.97 (0.01)	0.50 (0.09)/0.53 (0.02)	0.80 (0.32)/0.47 (0.20)	3.19 (2.30)/−0.29 (0.87)
16	0.06 (0.05)/0.98 (0.02)	0.06 (0.02)/0.98 (0.01)	3.28 (0.56)/−11.16 (8.54)	3.13 (0.69)/−10.55 (8.97)
17	2.18 (0.96)/0.71 (0.14)	1.24 (0.33)/0.65 (0.07)	6.09 (0.56)/−20.94 (9.45)	6.58 (0.76)/−17.29 (7.96)
**Subject-Wise Prediction Rates (Standard Deviation in Parentheses) for Arousal Detection.**				
**Subject**	**LSTM Raw: MSE/** $R^{2}$	**LSTM Base: MSE/** $R^{2}$	**AdaBoost Base: MSE/** $R^{2}$	**Random Forest Base: MSE/** $R^{2}$
2	1.48 (0.32)/−2.04 (1.19)	0.97 (0.11)/−9.65 (8.49)	0.89 (0.09)/−0.13 (0.25)	1.21 (0.38)/0.25 (0.30)
3	2.21 (0.71)/0.61 (0.09)	1.15 (0.41)/0.31 (0.21)	1.19 (0.10)/0.62 (0.03)	0.97 (0.11)/0.67 (0.01)
4	6.64 (0.92)/−1.73 (0.64)	0.45 (0.09)/0.88 (0.03)	0.67 (0.04)/0.87 (0.01)	0.63 (0.02)/0.86 (0.02)
5	0.35 (0.16)/0.95 (0.02)	0.67 (0.33)/0.80 (0.10)	0.69 (0.09)/0.81 (0.02)	30.45 (0.28)/0.87 (0.07)
6	0.10 (0.06)/−0.75 (0.13)	0.07 (0.03)/0.57 (0.20)	3.65 (0.04)/−0.08 (0.01)	3.09 (0.59)/−0.04 (0.04)
7	0.02 (0.02)/1.00 (0.03)	0.30 (0.08)/0.91 (0.03)	3.74 (0.07)/0.02 (0.01)	4.47 (0.76)/−0.78 (0.81)
8	0.18 (0.17)/0.94 (0.06)	0.04 (0.01)/0.99 (0.00)	1.82 (0.12)/−1.20 (0.28)	2.00 (0.29)/−1.92 (1.01)
9	0.18 (0.13)/0.71 (0.15)	0.19 (0.10)/0.73 (0.09)	2.29 (0.13)/0.20 (0.01)	1.39 (0.97)/0.39 (0.20)
10	0.14 (0.13)/0.99 (0.01)	0.08 (0.06)/0.98 (0.01)	1.71 (0.20)/0.41 (0.08)	2.48 (0.60)/−0.19 (0.54)
11	0.38 (0.64)/0.77 (0.39)	0.06 (0.03)/0.98 (0.01)	2.28 (0.30)/−0.79 (0.25)	1.94 (0.31)/−0.91 (0.34)
13	2.57 (2.11)/0.43 (0.58)	0.97 (0.26)/0.73 (0.07)	2.54 (0.38)/−0.53 (0.41)	1.39 (1.12)/0.24 (0.75)
14	1.62 (0.84)/0.69 (0.21)	1.75 (0.53)/0.44 (0.18)	5.21 (0.75)/−205.86 (339.33)	4.47 (1.25)/−208.87 (337.77)
15	0.22 (0.14)/0.87 (0.09)	0.28 (0.26)/0.87 (0.12)	1.24 (0.29)/0.53 (0.13)	2.77 (1.37)/−2.14 (2.60)
16	0.24 (0.14)/0.97 (0.02)	0.23 (0.15)/0.93 (0.04)	1.03 (0.0)/0.49 (0.0)	0.93 (0.10)/0.57 (0.08)
17	5.24 (1.57)/0.46 (0.23)	1.29 (0.64)/0.63 (0.19)	7.16 (0.31)/−22.30 (6.73)	5.70 (1.39)/−18.67 (8.54)

**Table 6 sensors-23-01598-t006:** Average recognition rates (standard deviation in parentheses) using LSTM regression model with baseline reduction with different sensor combinations.

Sensors	Valence: MSE/$R^{2}$	Arousal: MSE/$R^{2}$
EDA + ST + BVP	0.43 (0.05)/0.71 (0.04)	0.59 (0.06)/0.81 (0.02)
EDA + BVP	0.42 (0.05)/0.72 (0.05)	1.13 (0.10)/0.61 (0.03)
BVP + ST	0.72 (0.12)/0.52 (0.09)	0.92 (0.15)/0.71 (0.05)
EDA + ST	0.82 (0.24)/0.44 (0.18)	0.87 (0.11)/0.70 (0.04)
EDA	0.91 (0.08)/0.39 (0.05)	1.17 (0.22)/0.58 (0.08)
BVP	0.72 (0.07)/0.48 (0.05)	1.07 (0.08)/0.64 (0.03)
ST	1.76 (0.09)/−0.62 (0.08)	3.60 (0.27)/−0.70 (0.15)

**Table 7 sensors-23-01598-t007:** Results from different emotion classes when using features extracted from only some of the sensors (EDA and BVP for valence and EDA and ST for arousal) and LSTM regression model with baseline reduction.

Subject	Valence (EDA + BVP): MSE/$R^{2}$	Arousal (EDA + ST): MSE/$R^{2}$
2	0.61 (0.04)/0.00 (0.00)	0.69 (0.08)/−1.31 (2.56)
3	2.03 (0.41)/−0.55 (0.14)	0.69 (0.46)/0.62 (0.27)
4	0.16 (0.03)/0.85 (0.02)	0.36 (0.04)/0.91 (0.01)
5	0.15 (0.04)/0.79 (0.01)	0.19 (0.07)/0.94 (0.02)
6	0.04 (0.01)/0.95 (0.01)	0.14 (0.15)/0.79 (0.13)
7	0.04 (0.00)/0.83 (0.02)	0.44 (0.13)/0.86 (0.04)
8	0.02 (0.00)/0.97 (0.00)	0.08 (0.00)/0.97 (0.00)
9	0.28 (0.00)/0.39 (0.07)	0.04 (0.03)/0.86 (0.04)
10	0.11 (0.12)/0.96 (0.03)	0.05 (0.02)/0.98 (0.01)
11	0.09 (0.06)/0.94 (0.05)	0.18 (0.06)/0.94 (0.02)
13	0.81 (0.02)/0.36 (0.06)	2.26 (0.59)/−0.48 (1.05)
14	0.77 (0.04)/0.43 (0.08)	3.81 (0.96)/−1.14 (0.70)
15	0.38 (0.01)/0.59 (0.00)	0.07 (0.01)/0.96 (0.00)
16	0.10 (0.03)/0.96 (0.01)	0.56 (0.01)/0.82 (0.00)
17	0.23 (0.00)/0.93 (0.00)	1.12 (0.32)/0.68 (0.10)

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
