# Peer review of "Predicting Emotion with Biosignals: A Comparison of Classification and Regression Models for Estimating Valence and Arousal Level Using Wearable Sensors"

_sensors, 2023, doi:10.3390/s23031598_

Round 1

Reviewer 1 Report

Title: bad english!

Estimating Valance and Arousal Level to Recognize 3

Emotions from Wearable Sensor Data

Try maybe:

Emotions’ Valance and Arousal Recognition

From Wearable Sensor measurements

Abstract – badly written – try to divide it to few subsections – background, experiments, results, conclusion

Regard the methods – need to be better written

1.   Emphasize better why you use these features – common approach is to use many and then use feature selection

2.   Equations 1-3 are trivial – no needed – only to reference

3.   Please use not

Results

Figure 4 not clear – add there axes. Also, add there the true reference – to be compared to the estimated feelings

I miss also the separation of the emotions in PCA or TSNE  plan – this is MUST!

Need also to represent the classification/regression accuracy of all methods in one figure – better to do it also in classification form with ROC curve

The paper is not ready for publication - need to add more for analsyis part to be sure it works well. also to compare to other methods. 

Author Response

Thank you for the comments! Here is what I did based on them:

Title: bad english! 

Abstract – badly written – try to divide it to few subsections – background, experiments, results, conclusion

 Article now has a new title and abstract is rewritten.

Regard the methods – need to be better written

  1. Emphasize better why you use these features – common approach is to use many and then use feature selection
  2. Equations 1-3 are trivial – no needed – only to reference
  3. Please use not

Thank you for suggestions! We have now better emphasized why these features were used, and Equations 2-3 are removed. However, I left Equation 1 as normalization is one of the key elements of this work, and in Discussion it is explained why z-score is not a good option. 

Results

Figure 4 not clear – add there axes. Also, add there the true reference – to be compared to the estimated feelings

 This figure is updated, this comment made article much better.

I miss also the separation of the emotions in PCA or TSNE  plan – this is MUST!

 t-SNE figure added.

Need also to represent the classification/regression accuracy of all methods in one figure – better to do it also in classification form with ROC curve

What I did here was that I studied the best methods in more detail, table 4 now shows more detailed comparison of LSTM models. LSTM outperforms other models, so they we not studied in detail In addition, to Figures 5 and 6 I added results from LSTM based classification method as well, so the results of the models can be compered visually.

The paper is not ready for publication - need to add more for analsyis part to be sure it works well. also to compare to other methods. 

Discussion and Conclusions are completely rewritten, Much work has also been done with the Results section. Table added to related work to make comparison to previous works easier. In addition to modification related to your valuable comments above, a lot of work has been done to improve other parts of the article as well.

Reviewer 2 Report

Estimating Valance and Arousal Level to Recognize Emotions from Wearable Sensor Data

1. Starting from the introduction part, contributions are not clearly defined. The manuscript should reflect the research contributions in bullet points. The introduction does not reflect the noble contributions at all.

2. Section 2 related work section should reflect the key findings, and comparisons so require major reformation, only paragraphs do not seem fit here. In this section, some tables should be used for summarising the key points of the previously published work as compared with the current manuscript state.

3. In the result section, only two performance metrics are used, more performance metrics should be used to analyse the performance of the proposed research. For performance comparison, more performance parameters should be utilized.

4. Comparison of prediction models and normalization methods should be represented in a pictorial way with clarity of each step. Prediction models and normalization methods are the main building blocks for this research work so need a proper illustration with clear diagrams and phases etc.

5. One detailed flow chart should be included. In the proposed methodology section, a detailed flow chart should be included to highlight the key steps and details of the proposed research.

6. In section 5.2 subject wise results should be represented with tables/graphs etc.

7. Discussion section requires major revisions. Discussions should illustrate the key findings and justified reason but in the current state of the manuscript, it is completely missing.

8. Conclusion should reflect crystal clear findings rather than discussions about general outcomes. The conclusion should be re written with a discussion of the key findings only.

Overall lots of work should be carried out to enhance the quality of the manuscript. Therefore, the current state of the manuscript is not up to the mark for publication.

Author Response

Thank you for the comments! Here is what I did based on them:

  1. Starting from the introduction part, contributions are not clearly defined. The manuscript should reflect the research contributions in bullet points. The introduction does not reflect the noble contributions at all.

This is a valuable comment, thanks! Bullet points added.

  1. Section 2 related work section should reflect the key findings, and comparisons so require major reformation, only paragraphs do not seem fit here. In this section, some tables should be used for summarising the key points of the previously published work as compared with the current manuscript state.

Table added to make the comparison of previous works and our work easier.

  1. In the result section, only two performance metrics are used, more performance metrics should be used to analyse the performance of the proposed research. For performance comparison, more performance parameters should be utilized.

What I did here was that I studied the best methods in more detail, table 4 now shows more detailed comparison of LSTM models. LSTM outperforms other models, so they we not studied in detail In addition, to Figures 5 and 6 I added results from LSTM based classification method as well, so the results of the models can be compered visually. Figure 7 also improved to better show how well method works compared to true targets.

  1. Comparison of prediction models and normalization methods should be represented in a pictorial way with clarity of each step. Prediction models and normalization methods are the main building blocks for this research work so need a proper illustration with clear diagrams and phases etc.
  2. One detailed flow chart should be included. In the proposed methodology section, a detailed flow chart should be included to highlight the key steps and details of the proposed research.

Regarding comments 4 and 5, we have added Figures 3 and 4 to better explain how experiments were performed. Figure 3 shows how datasets to compare normalizations methods were created and Figure 4 focuses on showing how data was divided into training and testing.

  1. In section 5.2 subject wise results should be represented with tables/graphs etc.

These are in Table 5. Also Figures 5 and 6 show how subject wise results differ between classification and regression methods.

  1. Discussion section requires major revisions. Discussions should illustrate the key findings and justified reason but in the current state of the manuscript, it is completely missing.
  2. Conclusion should reflect crystal clear findings rather than discussions about general outcomes. The conclusion should be re written with a discussion of the key findings only.

Discussion and Conclusions are completely rewritten, Much work has also been done with the Results section.

Overall lots of work should be carried out to enhance the quality of the manuscript. Therefore, the current state of the manuscript is not up to the mark for publication.

In addition to modification related to your valuable comments above, a lot of work has been done to improve other parts of the article as well.

Reviewer 3 Report

In this article, the authors studied two types of prediction models, and different methods to normalize data are also compared. They further studied which biosignals and biosignal combinations provide the most reliable prediction models to detect valence and arousal levels.

This is an interesting study, however, can please authors add a couple of sentences on the novelty in the introduction?

Did the authors compare the proposed method with other existing methods in detail?

Author Response

Thank you for the comments! Here is what I did based on them:

In this article, the authors studied two types of prediction models, and different methods to normalize data are also compared. They further studied which biosignals and biosignal combinations provide the most reliable prediction models to detect valence and arousal levels.

This is an interesting study, however, can please authors add a couple of sentences on the novelty in the introduction?

This is a valuable comment, thanks! Bullet points added to Introduction to clarify the novelty of this work.

Did the authors compare the proposed method with other existing methods in detail?

Similar studies are not available. However, to make comparison to related work easier, we added Table to related work to make the comparison of previous works and our work easier.

Round 2

Reviewer 1 Report

Some nice improvements.

Still, english still not in level of presenting

I have difficult to understand why you don't give proper error anlasiys using classfication (of the two states) and give statistics of accruacy, sensetivity, specificity, etc. Instead you used MSE - which is not very defined. Also, you say you use new normalization, which is part of your contribution - need more support for the contribution of the normalization - a graph of results with regular normalization, and your claimed better one.  Also, some users have not great results - your conclusion need be rephrased

Author Response

Thank you for the comments! Here is what I did based on them:

Some nice improvements.

Thank you for these kind words!

Still, english still not in level of presenting

English has been improved.

I have difficult to understand why you don't give proper error anlasiys using classfication (of the two states) and give statistics of accruacy, sensetivity, specificity, etc. Instead you used MSE - which is not very defined.

Thank you for this comment, this has now clarified in the text: “Traditionally, the performance of the classification methods is analyzed using performance metrics such as accuracy, sensitivity, specificity, etc. However, as in this article, the idea is to compare classification and regression methods, and regression models cannot be analyzed using these metrics, both classification and regression models are analyzed using MSE, RMSE, R^2, and MAE. In fact, as valence and arousal are continuous phenomena and the targets for them are ordinal, it is natural that in this article all the models are analyzed using these metrics. Moreover, as both classification and regression models are evaluated using the same performance metrics, their comparison is easy”

As an example, let us assume that we are predicting the level of valence and the correct level is in this case 0. Now we have 2 classifiers, one predicts valence level as 1 and other as 4. If we are now using for instance accuracy or sensitivity to measure the performance of these models, both metrics consider these results equally wrong. However, as valence levels are ordinal, classifier which predicted the level as 1 was more correct that the one who predicted the level as 4. Now when we are measuring the performance using MSE, RMSE, R^2, or MAE they understand that model 1 performed better than model 2.

Also, you say you use new normalization, which is part of your contribution - need more support for the contribution of the normalization - a graph of results with regular normalization, and your claimed better one. 

This is good comment, normalization had too small role in results-section. We added there Figure 7 and text as well.

Also, some users have not great results - your conclusion need be rephrased

Sentence about this is added to conclusions.

Reviewer 2 Report

Estimating Valance and Arousal Level to Recognize Emotions from Wearable Sensor Data

One significant point should be included in the current state of the manuscript

1. Complexity analysis should be included in the manuscript.

Author Response

One significant point should be included in the current state of the manuscript

  1. Complexity analysis should be included in the manuscript.

Thank you for the comment! I added more details about the model to Section 4.2. However, I feel that detailed complexity analysis is out-of-the-scope of this article, as in this article we are just showing the it is possible to predict valence and arousal levels based on the data of wrist-worn sensor and the we are not yet trying to detect these in real-time using the calculation power of these devices. Thus, I added more detailed complexity analysis as a future work, now I just mention some details about the model to give understanding that the model is not too complex at least.

Round 3

Reviewer 1 Report

Better! Try to give more explenation why the t-sne data is seperated to different clusters, and how it imply to the results of the algorithm.

Figure 5,6 - better to have legend than to explain by words

Author Response

Better! Try to give more explenation why the t-sne data is seperated to different clusters, and how it imply to the results of the algorithm.

Thank you for encouraging words and valuable comments! More explanations related to T-SNE added, to both Discussion and Experimental dataset-sections.

Figure 5,6 - better to have legend than to explain by words

Legend added.